# Challenges in the Diagnosis of XY Differences of Sexual Development

**DOI:** 10.3390/medicina58121736

**Published:** 2022-11-27

**Authors:** Žana Bumbulienė, Diana Bužinskienė, Greta Banuškevičienė, Evelina Šidlovska, Eglė Preikšaitienė, Algirdas Utkus

**Affiliations:** 1Vilnius University, Faculty of Medicine, LT-03101 Vilnius, Lithuania; 2Centre of Obstetrics and Gynaecology, Institute of Clinical Medicine, Faculty of Medicine, Vilnius University, LT-08661 Vilnius, Lithuania; 3National Center of Pathology, Affiliate of Vilnius University Hospital Santaros Klinikos, LT-08406 Vilnius, Lithuania; 4Department of Human and Medical Genetics, Institute of Biomedical Sciences, Faculty of Medicine, Vilnius University, LT-08661 Vilnius, Lithuania

**Keywords:** primary amenorrhea, Swyer syndrome, complete gonadal dysgenesis, complete androgen insensitivity syndrome

## Abstract

*Background*: We report the clinical case of female patient with 46,XY difference of sexual development (DSD) and discuss the challenges in the differential diagnosis between complete gonadal dysgenesis (also called Swyer syndrome) and complete androgen insensitivity syndrome. *Case Presentation*: The patient’s with primary amenorrhea gynaecological examination and magnetic resonance imaging (MRI) revealed the absence of the uterus and a very short vagina. Two sclerotic structures, similar to ovaries, were recognised bilaterally in the iliac regions. Hormonal assay tests revealed hypergonadotropic hypogonadism and the testosterone level was above normal. The karyotype was 46,XY and a diagnosis of Swyer syndrome was made. At the age of 41, the patient underwent a gynaecological review and after evaluating her tests and medical history, the previous diagnosis was questioned. Therefore, a molecular analysis of sex-determining region Y (SRY) and androgen receptor (AR) genes was made and the results instead led to a definite diagnosis of complete androgen insensitivity syndrome. *Conclusions*: The presented case illustrates that differentiating between complete gonadal dysgenesis and complete androgen insensitivity can be challenging. A well-established diagnosis is crucial because the risk of malignancy is different in those two syndromes, as well as the timing and importance of gonadectomy.

## 1. Introduction

Differences/disorders of sexual development (DSD) encompass a group of congenital conditions characterised by atypical development of chromosomal, gonadal and phenotypic sex [1]. The terminology of different sex development is currently preferred. DSD is divided into three groups according to the chromosomal component: 46,XX DSD; 46,XY DSD and sex chromosomal DSD [1].

46,XY DSD is a group of rare and challenging conditions that can be caused by abnormalities of karyotype, gonadal formation, androgen synthesis and androgen action [2]. The diagnosis of patients with 46,XY DSD is mainly clinical and is usually identified during an investigation for primary amenorrhea or delayed puberty. For DSD patients having Y chromosomes, prophylactic gonadectomy should be considered due to the increased risk of gonadal malignancy, but in different conditions, this risk may vary, so it is important to make the right diagnosis for a patient. The most frequent causes of 46,XY females are androgen insensitivity syndrome (AIS) and gonadal dysgenesis [3].

Androgen insensitivity syndrome can be complete, partial or mild. Depending on the level of androgen insensitivity, an affected person’s sex characteristics can vary from mostly female to mostly male. Complete androgen insensitivity syndrome (CAIS) is the most common type and affects 2–5:100,000 females [4]. Although the literature on androgen insensitivity syndrome seems extensive, the question remains relevant, judging by the fact that, in the scientific literature during 2022, there are published 22 articles on complete androgen insensitivity syndrome (of which, five were case reports) and 8 articles on partial androgen insensitivity syndrome (of which, one was a case report). CAIS is caused by pathogenic variants in the AR gene and is manifest as a complete absence of androgen receptors functions and, consequently, no androgenic action on target cells. CAIS clinical findings are the presence of a short vagina, the absence of Müllerian structures and normally developed undescended testes, usually leading to the formation of either unilateral or bilateral inguinal hernias. Phenotypically, breasts and female adiposity are developed normally, but pubic and axillary hair is absent or very rare [5]. The risk of malignancy in cases of CAIS is low. In the scientific literature, it varies from 1 to 3–5% but increases to 15% after puberty, so gonadectomy can be deferred until early adulthood [6].

Complete gonadal dysgenesis, also known as Swyer syndrome, typically develops due to gene mutations in the SRY sex-determining region of the Y chromosome, but it can be caused by other Testis Determining Factors gene pathogenic variants as well. The frequency is approximately 1:80,000 people [7]. Phenotypically patients have eunuchoid body proportions, and there are no signs of sexual development because of non-functional (streak) gonads. The lack of anti-Müllerian hormone (AMH) results in a lack of regression of Müllerian duct structures, including the uterus, fallopian tubes, and upper 1/3 of the vagina (i.e., they are present when they would be expected to be absent). However, due to decreased estrogen production, the uterus may be small and not be identified during the initial assessment. Thus, the differentiation of these syndromes can be challenging. The residual gonadal tissue often becomes cancerous: the risk of malignancy can be as high as 40% and occurs in childhood or early adolescence, and even before the condition is suspected, so that gonadectomy is recommended as soon as complete gonadal dysgenesis is diagnosed [8].

## 2. Case Report

A 16-year-old female was referred to her gynecologist with primary amenorrhea in 1994. A gynecological examination revealed the congenital absence of the uterus and internal genitals, and a very short vagina was also observed. From her past medical history, it is known that her neonatal weight and height were 3500 g and 53 cm, respectively. In addition, inguinal hernias were operated on during her childhood. During that consultation, no genetic tests were performed. The working diagnosis was primary amenorrhea and uterine aplasia. The patient did not apply later, and the following 14 years of the patient’s medical history are unknown. 

At the age of 30, the patient consulted an endocrinologist. The examination findings were as follows: the patient was 178 cm tall and weighed 68 kg, her secondary sexual characteristics were poorly developed and her breasts were small. Her hormonal assay tests revealed hypergonadotropic hypogonadism: follicle-stimulating hormone (FSH) 42.6 U/I (normal range male 0.9–11.9, female 0.9–9.3 U/I), luteinizing hormone (LH) 44.1 U/I (normal range male 1.1–8.7, female 0.9–9.3 U/I), estradiol 118 pmol/L (normal range male 40.4–161.5, female 77–1145 pmol/L). Testosterone levels were elevated at 17.2 nmol/L (normal range male 5.8–30.4, female 0.3–4.5 nmol/L), while prolactin levels and thyroid-stimulating hormone were normal. The patient was referred for genetic counselling. 

Karyotyping revealed 46,XY. Pelvic magnetic resonance imaging (MRI) showed an underdeveloped uterus with the ovaries not clearly visible. Two sclerotic structures, similar to ovaries, measuring 16 × 11 × 28 mm and 18 × 11 × 18 mm with a few small cystic inclusions were recognised in both the left and right iliac regions, medially to the external iliac arteries. The diagnosis was unclear; therefore, an endocrinological consultation was held. Due to the high gonadotropin levels, the diagnosis of complete gonadal dysgenesis was confirmed, and estrogen replacement therapy was started. 

After 11 years of treatment, the MRI scan was repeated: the uterus was not detected, the vagina was 3 cm long, and the sclerotic structures were still visible in both iliac regions remaining the same size. After evaluating the patient’s external genitalia (female external genitalia without virilisation), hormonal tests (male-range testosterone level), imaging (no uterus) and medical history (inguinal hernias during childhood), the gynaecologist questioned the previous Swyer syndrome diagnosis and suspected a complete androgen insensitivity syndrome. 

To confirm the diagnosis at the age of 41 years, a molecular analysis of SRY and AR genes was performed. The hemizygous variant NM_000044:c.1706G>A, NP_000035:p.(Gly569Glu) of *AR* gene was detected. The variant has been reported in the Human GeneMutation Database in association with androgen insensitivity syndrome [9]. Additionally, it was not found in the Genome Aggregation Database (gnomAD) exomes and was predicted to be pathogenic using bioinformatics tools. Therefore, it has been classified as pathogenic. The patient was diagnosed with complete androgen insensitivity syndrome.

The molecular diagnosis of CAIS was also confirmed for the niece of the proband (Figure 1, III-2). Her karyotype was 46,XY, and the familial mutation in the *AR* gene was detected. The sister of the proband (II-1) was a carrier of the mutation. 

After confirming the CAIS diagnosis patient continued estrogen replacement therapy, and it was recommended to remove the gonads. After a year, a prophylactic laparoscopic bilateral gonadectomy was performed. Intraoperatively a small amount of free fluid (about 20 mL) was found in the peritoneal cavity. There were no uterus and fallopian tubes (not formed). The liver, spleen, and intestines were without objective changes, and adhesions in the peritoneal cavity between the appendix and the anterior abdominal wall after a previous appendectomy were observed. Structures similar to ovaries were observed bilaterally medial to the external iliac arteries and were fixated to the abdominal wall with peritoneal folds and a round-shaped ligament (probably derived from the gubernaculum) on the posterior end of the structure. On the right side, there was an ovarian-like structure measuring 3 × 1.5 × 1 cm, and there was a 1.5 cm long structure next to its posterior end that resembled a part of a fallopian tube and was also fixated to the abdominal wall by peritoneal folds. On the left side, the ovarian-like structure was smaller (1.5 × 1 × 1 cm) and had no observable attachments to it (Figure 2). Both gonads were removed with a bipolar coagulator and scissors. Adhesions in the peritoneal cavity were removed as well. There were no complications during and after the operation.

The histological analysis of the patient’s removed gonads revealed that there were hypoplastic testicles, which is a clinical sign of androgen insensitivity syndrome (Figure 3, Figure 4, Figure 5 and Figure 6). After the operation, testosterone analysis was repeated-hormone level significantly decreased to 0.83 nmol/L, which is a normal female-range value for a woman at that age.

## 3. Discussion

This case demonstrates the difficulties in differentiating DSD because these conditions may manifest very similarly to each other. 

Hypergonadotropic hypogonadism is more common in Swyer syndrome cases because these patients have non-functional streak gonads, but the pituitary gland still stimulates them by releasing gonadotropin hormones. In this case, hypergonadotropic hypogonadism and some poorly developed secondary sex characteristics first led to the most probable diagnosis of complete gonadal dysgenesis. However, a high testosterone level, the absence of the uterus, the hypoplastic vagina and inguinal hernias in the past medical history were inconsistent with Swyer syndrome. These clinical findings are more typical for androgen insensitivity syndrome. On the other hand, hypergonadotropic hypogonadism is not typical in CAIS. However, in the adult patient with CAIS, the degeneration of the germ cells over time can be detected, so elevated gonadotropin levels might occur [10].

Usually, patients with Swyer syndrome have normally developed uterus, but since it has not been stimulated by estrogens yet, it may not be identified by imaging techniques. Therefore, the clinician’s experience plays an important role in such cases, and it may even be necessary to delay the final diagnosis of uterine agenesis and similar urogenital anomalies until after puberty [11]. That is why complete gonadal dysgenesis may be misdiagnosed sometimes.

Another noticeable difference is a hormonal interpretation of these two syndromes. In Swyer syndrome, testosterone levels should not be elevated and ought to be of female-range value. In CAIS cases, the situation is the opposite—typically, a testosterone level is elevated. The man-range testosterone level in women would be expected to cause virilisation and clitoromegaly unless there is a defect in the androgen receptor, and CAIS should have been strongly suspected based on this. Our patient had typically female-appearing external genitalia, which would support a CAIS diagnosis in light of hyperandrogenemia. 

High risk of misdiagnosis in patients with primary amenorrhea was mentioned by Porsius et al. In determining possible 46,XY DSD condition they suggested to test not only testosterone, but also AMH [12].

Only the molecular analysis of SRY and AR genes clarified the diagnosis in this ambiguous situation. The found mutation in the AR gene led to a complete androgen insensitivity syndrome diagnosis (Table 1). In addition, after surgery, the diagnosis was confirmed by histological analysis of removed gonads (hypoplastic testicles).

It is important to differentiate these two syndromes because the risks of malignancy and recommendations for the timing of gonadectomy depend on the diagnosis. In Swyer syndrome, the risk of malignancy is high and can develop in childhood or early adolescence. Surgical removal of gonads is recommended immediately after the diagnosis [8]. While in CAIS the risk of malignancy is low (vary from 1% to 3–5%) and mostly develops after puberty (risk grows to 15%), therefore gonadectomy is recommended in early adulthood, or recently, adult patients often refuse to remove the gonads. In both conditions, hormonal replacement therapy is indicated.

While it is necessary to know specific clinical findings for different conditions, it is also helpful to remember that there always can be exceptions. The authors recommend careful consideration of all testing methods and, in the presence of an uncertain diagnosis, the use of molecular analysis of genes to avoid misinterpretation and misdiagnosis.

## 4. Conclusions

Differentiating complete gonadal dysgenesis and complete androgen insensitivity can be challenging because these syndromes can manifest in a similar manner. It is crucial to differentiate these two syndromes because the risks of malignancy and recommendations for gonadectomy depend on the diagnosis. Molecular analysis can help reveal the actual diagnosis for similar uncertain clinical cases.

## Figures and Tables

**Figure 1 medicina-58-01736-f001:**
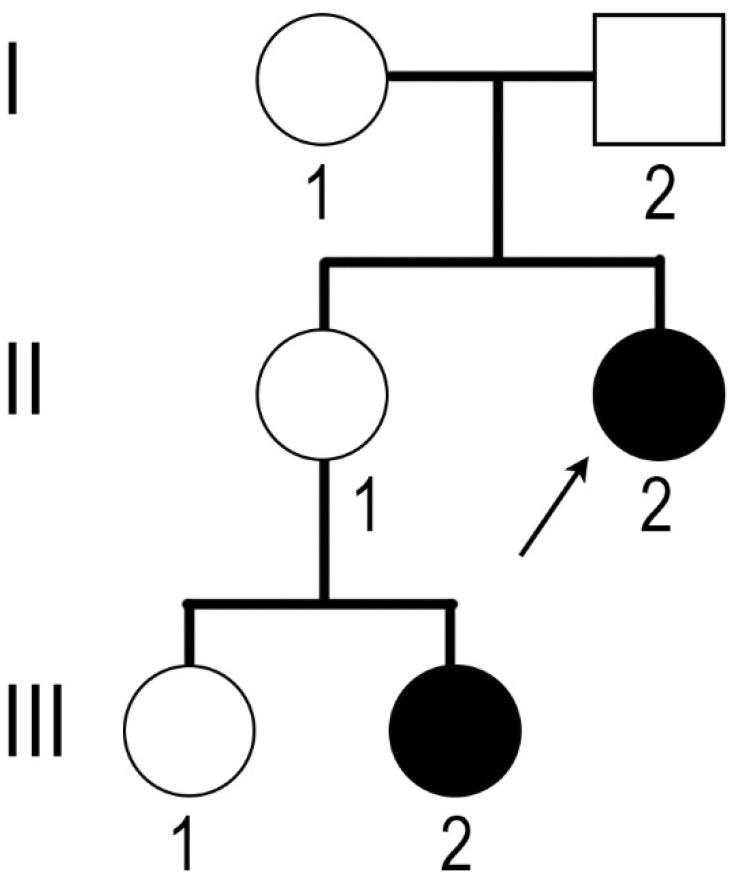
Genealogy of the family. Black symbols denote individuals with clinical features of complete androgen insensitivity syndrome (CAIS). I–III denote the generations; numbers 1, 2 reflect the order of birth in the family; arrow marks the patient discribed in the case report.

**Figure 2 medicina-58-01736-f002:**
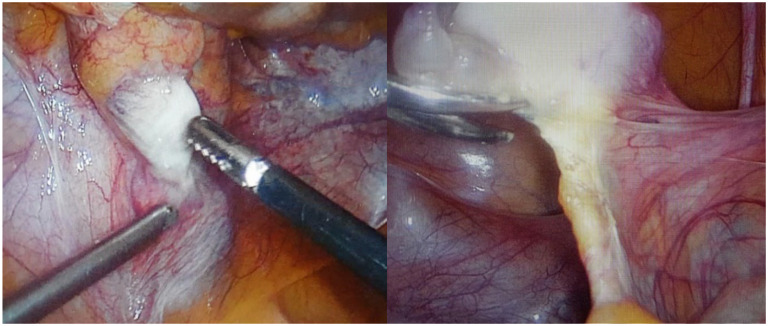
Intraoperative photography of the **left** and **right** gonads.

**Figure 3 medicina-58-01736-f003:**
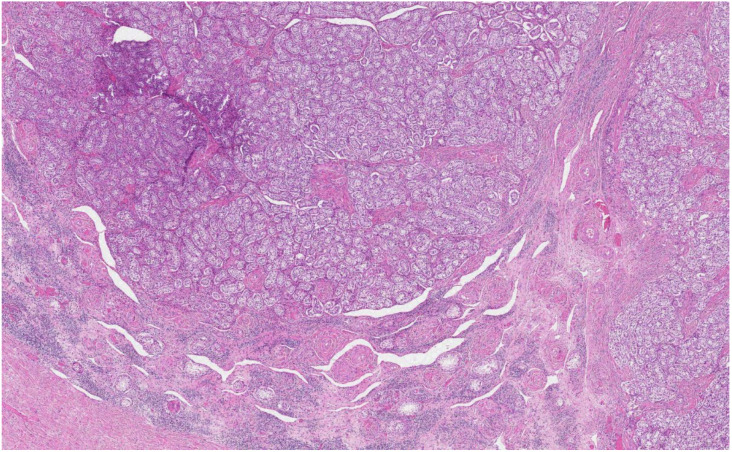
Microscopic examination revealed testes, formed by small tubules with mostly absent lumen, composed of Sertoli cells surrounded by fibrous stroma, or stroma resembling ovarian stroma, between tubules and separately at the periphery of tubules-nodules and small nests of Leydig cells with abundant eosinophilic cytoplasm and round nuclei (H&E staining, ×30).

**Figure 4 medicina-58-01736-f004:**
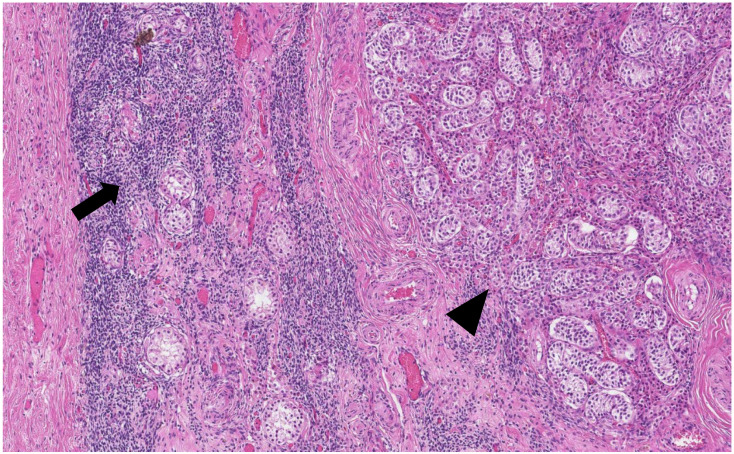
Ovarian-type stroma (arrow) and tubules composed of immature Sertoli cells separated by Leydig cells (arrowhead) (H&E staining, ×50).

**Figure 5 medicina-58-01736-f005:**
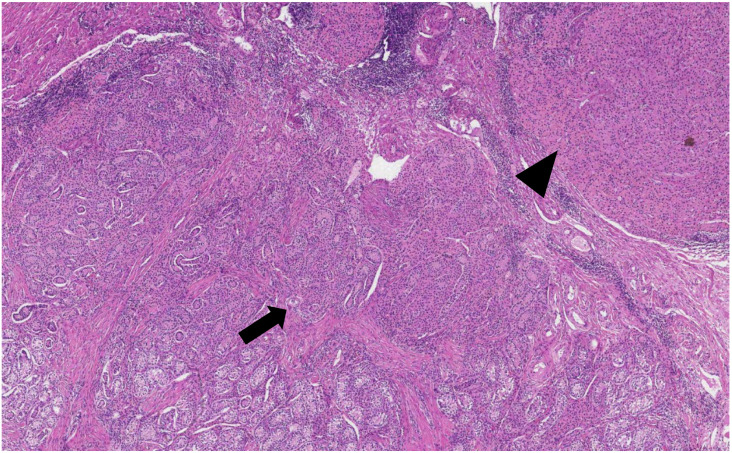
Small tubules composed of immature Sertoli cells separated by Leydig cells (arrow) and nodular Leydig cell hyperplasia (arrowhead) (H&E staining, ×50).

**Figure 6 medicina-58-01736-f006:**
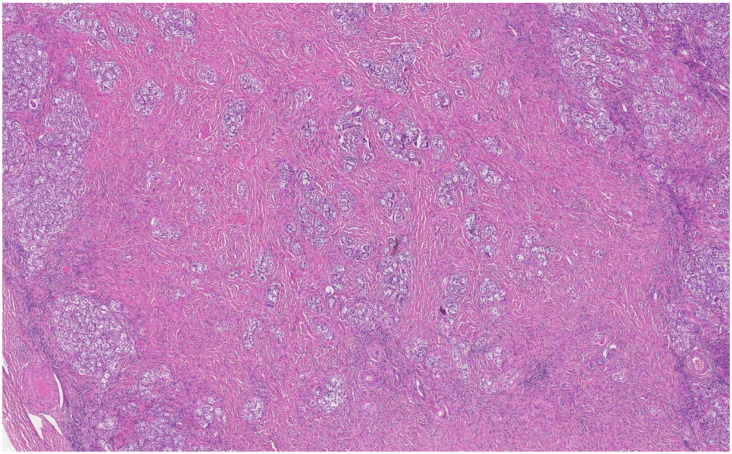
One of the testes revealed a nodule composed of spindle cells, abundant collagen fibers and tubular structures lined by Sertoli cells (H&E staining, ×40).

**Table 1 medicina-58-01736-t001:** Main characteristics of complete gonadal dysgenesis and complete androgen insensitivity syndrome compared to our case report patient’s clinical findings.

	Swyer s.-Complete Gonadal Dysgenesis	Complete Androgen Insensitivity Syndrome	Clinical Case
Secondary sexual characteristics	Eunuchoid body proportions, no signs of sexual development	Breasts and female adiposity develop normally, but pubic and axillary hair is absent or very rare	Secondary sex characteristics poorly developed, small breasts
Uterus	Normally developed	Absent	Absent
Gonads	Non-functional “streak” gonads	Normally developed undescended testes	Hypoplastic undescended testicles
Vagina	Normally developed	Short vagina	Short vagina
Gonadotropins	Elevated	Normal	Elevated
Testosterone	Female-range	Male-range	Male-range

## Data Availability

Not applicable.

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
