# Peer review of "Challenges in the Diagnosis of XY Differences of Sexual Development"

_medicina, 2022, doi:10.3390/medicina58121736_

Round 1

Reviewer 1 Report

I think manuscript wroted by Žana Bumbulienė and colleagues need minor revision, especially  Ethical Statement "Institutional Review Board Statement: Not applicable???. "- It was genetic counselling, blood tests, you need retrospectively to receive.

Manuscript contain nice table and graphic figure.

Please add, how many cases  CAIS, PAIS was described to 2022 in literature- update.

Thank you

Kinds regards,

Izabela Winkler

Author Response

Point 1: I think manuscript wroted by Žana Bumbulienė and colleagues need minor revision, especially  Ethical Statement "Institutional Review Board Statement: Not applicable???. "- It was genetic counselling, blood tests, you need retrospectively to receive.

Point 1: The study was conducted according to the guidelines of the Declaration of Helsinki, and approved by the Ethics Committee of Vilnius university hospital Santara Clinic. We haves permissions from both the patient and the  Vilnius university hospital Santara Clinics to publish the case. It was a mistake that we wrote ethical statement 'not applicable', we will definitely change that. We can provide supporting documents.

Point 2: Please add, how many cases  CAIS, PAIS was described to 2022 in literature- update.

Point 2:  Although there seems to be a lot of literature on androgen insensitivity syndrome, the question remains relevant, judging by the fact that in scientific literature during the 2022 year there are published 22 articles on complete androgen insensitivity syndrome (of which 5 were case reports) and 8 articles on partial androgen insensitivity syndrome (of which 1 was case report).

Reviewer 2 Report

I would like to congratulate the authors for managing this case. The management of cases with DSD is very difficult, and often requires a multidisciplinary approach.

I am f the opinion that the misdiagnosis of Swyer syndrome could have been avoided upfront. My detailed comments:

This was a case of CAIS, misdiagnosed as Swyer syndrome due to incomplete clinical history and examination and wrong interpretation of tests.   1. The patient had history of inguinal hernias suggestive of undescended gonads.   2. MRI showed presence of uterus Initially but later on no uterus-- which is a stark finding. The interpretation of the tests might be wrong initially as later investigations all showed no uterus.   3. Normal range testosterone levels also favour CAIS over complete gonadal dysgenesis.   4. Increased levels of gonadotrophins also can be seen in CAIS.   Therefore, careful evaluation is necessary to identify the correct diagnosis. Overall, this manuscript contributes very little to the already available literature.

Author Response

We agree with the assessment, but it again emphasizes the importance of detailed anamnesis and physical examination in such cases.

Reviewer 3 Report

The author presented an interesting case. It is essential to make the difference between XY Disorders of Sexual Development because the risk of malignancy is different. Therefore the management of the patient must be other. The topic of the article is with significant implications for medical practice. The subject is relevant and exciting to the field of the journal. The report makes a substantial contribution to the field. The text is clear and easy to read. The case has an excellent medical description. The overall paper is organized and well-written. The Discussions section has insightful and informative literature reviews. The conclusion is clear.

I congratulate all the authors for their efforts.

I have only a few remarks to make.

The Instructions for authors specify that Acronyms/Abbreviations/Initialisms should be defined the first time they appear in each of three sections: the abstract, the main text, and the first figure or table. Please check this aspect.

Author Response

Thank you for the review and comments. We will definitely change these aspects.

Round 2

Reviewer 2 Report

I would like to congratulate the authors for the management of this case. However, as explained previously, the misdiagnosis of Swyer syndrome was due to incomplete clinical history and examination, and wrong interpretation of tests. Therefore, a careful evaluation is necessary to identify the correct diagnosis. Overall, this manuscript contributes very little to the already available literature.